# Understanding age at menarche: Environmental and demographic influences over a quarter century in India

MD. Nahid Hassan Nishan[1]*, Anika Ferdous[2], M. Z. E. M. Naser Uddin Ahmed[1], Khadiza Akter[1,3]

**1** Department of Public Health, North South University, Dhaka, Bangladesh, **2** Department of Audiology and Speech-Language Pathology, Bangladesh University of Professionals, Dhaka, Bangladesh, **3** Department of Nursing, International University of Business Agriculture and Technology, Dhaka, Bangladesh

* nissan0808@yahoo.com

## Abstract

This study investigates the factors influencing the age of menarche in various Indian states over a quarter century from 1992 to 2019, with the aim of understanding how climate change and demographic factors have shaped menarche timing. Data from the Indian Demographic and Health Survey (DHS) for 1992–93 and 2019–21, along with climate data from NASA's POWER project, were analyzed using a cross-sectional study design, including 23,083 respondents from 1992 and 45,329 from 2019. Across most states, a slight decrease in age at menarche was observed, with the exception of Maharashtra, which showed an increase. Higher specific humidity was associated with earlier onset of menarche, whereas higher temperatures correlated with delayed onset. Improvements in educational attainment, particularly higher education levels, were strongly linked to earlier menarche, indicating that demographic changes had a significant influence. The findings highlight the need for public health interventions that improve nutrition, healthcare access, and educational programs to promote health awareness. Ongoing monitoring of climatic impacts on health is essential for understanding and mitigating the effects of environmental changes on menarche timing.

## Introduction

The onset of menstruation represents a significant biological and psychological transition, marking a girl's progression from childhood into adolescence. With the onset of menarche, females start their reproductive journey, referring to the first menstrual cycle that enables women to develop fertility through ovulation [1–3]. The first menstrual bleeding might not start at the same age for all girls. In general, menarche typically occurs between the ages of 12 and 13 globally, though this may vary.

**Data availability statement:** The raw datasets used in this study are publicly available from established data repositories. The Demographic and Health Surveys (DHS) program provides access to survey data upon registration at https://dhsprogram.com, and climate data were obtained from the NASA POWER Project at https://power.larc.nasa.gov/. Both sources are publicly accessible and subject to their respective data access policies.

**Funding:** The author(s) received no specific funding for this work.

**Competing interests:** The authors have declared that no competing interests exist.

Early menarche is generally defined as onset before age 12, while late menarche is considered after age 15. In India, the mean menarche is 12.77 years [4]. Early menarche has been associated with heightened risks of early sexual debut, adolescent pregnancy, polycystic ovarian syndrome, reproductive cancers, obesity, and mental health issues including depression and anxiety. In contrast, late menarche can signal undernutrition, delayed growth, or chronic illness, and has been linked to lower bone mineral density, dysmenorrhea, and cardiometabolic complications in later life [5–7]. Understanding the determinants of menarche timing is critical given its link to later reproductive and health outcomes.

the menarche change is changing across the globe. Especially in the second half of the 20th century, the age of menarche has decreased [8]. A study showed there is a significant variation in the age of first menstruation between urban and rural adolescents [5]. Throughout India, the average age of the first menstruation period in rural areas has been found to be 13.36 years [9]. Prior research also indicates variation in age at menarche across ethnic and caste groups in India, though findings are mixed. Some studies report earlier menarche among disadvantaged castes, while others suggest negligible differences among groups such as Brahmins and Rajput's [10,11]. Additionally, research shows that educated girls experience menarche earlier than illiterate girls [8]. Interestingly, daughters of educated parents have also been found to experience menarche earlier than their counterparts [7,12]. Religion may also play a role in shaping the timing of menarche through its influence on lifestyle, dietary habits, and cultural practices. For example, religious fasting, food taboos, and modesty norms can impact nutritional status, physical activity, and health-seeking behavior—all of which may affect pubertal development [13]. However, existing research on the direct physiological impact of religion on menarche remains limited and warrants further exploration.

Furthermore, climate could impact the timing of menarche in several ways [14,15]. These usually include temperature, humidity, and lighting. Studies found that menarche occurs more commonly in winter than in summer, and there is a late maturation among people who live in higher altitudes [16,17]. In addition, weather can trigger the nutritional status of individuals by altering food availability and consumption; thus, these factors can influence the age of first menstruation [15]. One proposed theory suggests that climate influences the timing of menarche through its effects on the body's neuroendocrine system. Specifically, reduced sunlight exposure—common in colder seasons or high-latitude regions—may increase melatonin secretion, which in turn can suppress the hypothalamic–pituitary–gonadal (HPG) axis that regulates puberty [15,18]. Additionally, colder temperatures and high-altitude environments elevate metabolic demands, potentially delaying the accumulation of body fat required to trigger puberty-related hormonal signaling [19]. This theory highlights how climatic factors can biologically mediate variations in menarcheal timing through hormonal and energy balance pathways.

However, though several research studies have been conducted on different aspects of menarche in India, there is still a research gap in finding the factors influencing the change in the age shift of menarche from the past to the present. There remains a novelty in this study as this will contribute to the further enrichment of the reproductive health of Indian adolescents. Therefore, this study aims to understand how environmental and

demographic factors influenced menarche timing over the last quarter century. This study will provide valuable insight regarding the trend shift of menarche in India over the quarter-century period. The findings are expected to inform policies and programs targeting adolescent development and climate-sensitive health interventions in India.

## Methodology

### Study description

We used India's Demographic and Health Survey (DHS) dataset, which contains nationally representative health and demographic data. Among them, according to the World Bank classification, India is categorized as a lower-middle-income country. There are several years of standard Indian DHS datasets suitable for this research. Still, for this study, we kept an interest in the Indian 1992–93 [20] dataset and the Indian 2019–21 [21] DHS dataset. Since we wanted to check the quarter-century change, we selected data from 1992 and 2019, as these two years contain all the necessary data and have more samples available. Moreover, we have taken NASA's (National Aeronautics and Space Administration) power project data to collect climate data. We have also taken data from 1992 and 2019 from them to make the model more suitable for analysis. We selected these states of India, which include Andhra Pradesh, Assam, Goa, Himachal Pradesh, Karnataka, Kerala, Maharashtra, Rajasthan, Uttar Pradesh, and West Bengal, as we made sure all necessary variables and data were available on both the 1992 and 2019 datasets.

### Study design

This study employed a cross-sectional design, which allows for the analysis of associations between variables but does not permit causal inferences. The objective was to explore the relationships between environmental and demographic factors and the timing of menarche. We used data from DHS and the NASA power project; all the data we used were secondary data. We used all the data that demonstrated the necessity of the research. Since we took data from DHS and NASA power projects, it is crucial to know their data collection technique. DHS's data collection method is a two-stage sampling design technique. Their Program has implemented a methodology to guarantee that health-related data can be compared across countries. This is achieved by using questionnaires that have been developed and refined over six program phases. While countries are encouraged to adopt the questionnaire, they also have the flexibility to include questions or exclude irrelevant ones. In addition, these surveys are conducted on a measurable scale, representing the population with sample sizes ranging from 5,000–30,000 households. The main questionnaires used are the Household, Women's, and Men's Questionnaires. Additional modules are added based on the interests of each country. The careful management of questionnaire changes, consistency of variables, and consideration of dropped questions contribute to the integrity of the collected data. This approach enables comparisons between phases and countries while ensuring the reliability of health-related data collected by the DHS Program.

On the other hand, NASA's Earth Science Division leads the POWER project in another field, which collects solar and meteorological data using satellite observations and assimilation models. Initially focused on energy concerns as an extension of the SSE project, it has since expanded its scope to include datasets on sustainable buildings and agroclimatology. The recent version, POWER Release, includes satellite data to monitor activity and uses the Modern Era Retrospective Analysis for Research and Applications (MERRA-2) assimilation model to analyze meteorological information. The information available includes spatial resolutions that correspond to the input data. These resolutions demonstrate progress by offering data for parameters. NASA's dedication to accuracy and dependability is evident in its inclusion of validation measures and uncertainty estimates based on surface measurements.

### Study participant

Since we relied on the DHS dataset for women's data, we chose the Individual (IR) file. In that, we first identified the years of data as 1992–93 contains both year samples, and 2019–21 has a three-year sample. During that year, DHS collected

data from multiple states of India, but not the same state over the period, which introduces a bias. For the reliability of the sample we used, we first filtered out the states' names from each year of the dataset we used. We identified states in which all the necessary data were available and present on both the 1992 and 2019 datasets. We already mentioned 10 states that we had selected for our study above. We again filtered individuals who provided their first menstruation period year among those states. Upon filtering them, we kept 23,083 samples from 1992 and 45,329 samples from the 2019 dataset. As for collecting the climate data, after identifying all the states, we had to collect their longitude and latitudes. For this, we used this [22] to note down the longitude and latitude of each selected state of India. After that, we used them in the NASA power project to filter out the climate data. We downloaded only the 1992 and 2019 datasets from them, filtered out the necessary variables that we used, and merged them with DHS to find the insight that we were looking for.

## Inclusion and exclusion criteria

Our inclusion criteria included women who gave information on their first menstruation period. Also, these women were chosen based on the selected states that we first identified. Any women who didn't provide that data or kept any variables unfilled were excluded from the study. Also, any data related to years other than 1992 and 2019 were excluded to keep the study more valid and reliable. On the other hand, in terms of NASA data, we filtered similarly based on the states we chose, kept only the necessary independent variable, and filtered out others. In both datasets, any sample which has missing values was totally excluded. While climate extremes may influence biological processes, the study utilized annual averages from NASA's POWER database to align with DHS survey timing and ensure geographic consistency. Seasonal or monthly climate data were not used, as the DHS does not capture month-specific menarche timing, making temporal alignment with short-term environmental exposures unfeasible. Additionally, age at menarche was self-reported, introducing the possibility of recall bias—particularly among older respondents recalling events from adolescence. However, such recall is typically accurate within a one-year margin. Any misreporting is likely nondifferential and would bias associations toward the null, but self-reported menarche remains the most practical and accepted measure in large-scale demographic research.

## Outcome measures

The outcome variable of this study was the women who provided information on the age of their first menstruation period. The DHS of India collects data from women aged 15–24 years, asking, "*How old were you when you had your first monthly period?*" This retrospective self-reported measure of age at menarche enables the analysis of changes over time and across demographic groups. The variable was kept in a continuous nature, and we took this from both the 1992 and 2019 datasets. After filtering out inclusion and exclusion criteria, we concluded the outcome variable for the samples.

## Independent variables

In this study, we identified several participants' demographics, who were given their data and selected based on both inclusion and exclusion criteria. Each of the respondents was categorized as urban or rural. The respondents' education levels were categorized as No Education, Primary, Secondary, and Higher. Since there are many religious beliefs in India in different states, the religious status of the individual is also categorized as Hindu, Muslim, Christian, Sikh, and Others. Religion was included as a predictor variable due to its potential influence on social and cultural practices, including gender norms, dietary restrictions, reproductive health attitudes, and access to care. Previous studies have noted that religious affiliation can affect pubertal development through pathways such as nutritional behavior, body perception, or stigma related to reproductive health [23,24]. In the Indian context, religion is also a strong marker of household structure, educational attainment, and access to adolescent health services, making it a relevant variable for understanding variations in menarche timing [25]. India also has many Ethnicities, which we categorized into Caste, tribe, and others. All of them are selected from the DHS alone. We kept the annual temperature at 2 meters and specific humidity at 2 meters

from the NASA power project dataset for climate data, as it best represents human environmental exposure, minimizing surface variability and ensuring consistency with prior health and climate studies. This standard measurement enhances the accuracy of assessing climatic influences on menarche timing. We can also choose from other available data, but the justification for taking this variable is that high temperatures can cause menstrual discomfort and lead to health problems related to heat, dehydration, and tiredness, which may pose significant reasons for changing menstrual age shifts. On the other hand, the specific humidity level plays a role in how we feel the heat, so staying adequately hydrated during menstruation is essential. Both variables were kept continuous in nature.

## Data analysis

All analyses were conducted using Stata version 17. The study combined two main data sources: the Demographic and Health Surveys (DHS) from India for the years 1992–93 and 2019–21, and NASA POWER climate datasets containing annual estimates of environmental indicators such as temperature and specific humidity.

## Data preparation and integration

DHS data underwent extensive cleaning to ensure reliability, particularly for variables like age at menarche, age at interview, and state-level geographic identifiers. Respondents' reported age at menarche was used in conjunction with their age at the time of survey to back-calculate the approximate year of menarche. Climate data were matched to each respondent based on the estimated year of menarche, calculated by subtracting the age at menarche from the year of the DHS survey. For example, a respondent surveyed in 2019 who reported menarche at age 14 was assigned climate data from 2005. This method allowed for temporal alignment between climate exposure and the onset of menarche.

The NASA POWER data, which are provided as gridded spatial datasets, were processed to extract annual mean temperature and specific humidity at the state level. DHS cluster GPS coordinates were used to spatially match each cluster to the nearest grid cells of climate data. While we recognize that pubertal timing may be influenced by cumulative or multi-year environmental exposures, we used the best available year-specific climate data due to limitations in historical climate resolution at the subnational level. Averaging climate variables across distant time points (e.g., between 1992 and 2019) would obscure meaningful exposure variation and introduce misclassification bias. Therefore, we opted for a year-specific matching approach to preserve the temporal integrity of the exposure-outcome relationship. We acknowledge this as a methodological limitation and recommend that future studies consider integrating multi-year or rolling average climate indicators where more granular environmental data are available. Moreover, annual climate estimates were averaged at the state level and merged with the DHS dataset using state-year identifiers. This process ensured harmonization across both data sources, enabling meaningful environmental comparisons.

Only states with complete DHS and NASA data across both time points were included in the analysis. Some northern states, such as Punjab and Jammu & Kashmir, were excluded due to incomplete records, which may limit the generalizability of findings at the national level. However, the included states represented broad geographic and climatic diversity, preserving analytical strength and regional insight.

## Descriptive and bivariate analyses

Initial analyses involved descriptive statistics to summarize sociodemographic and reproductive characteristics across states and time points. T-tests were used to assess differences in continuous variables across binary categories, while one-way ANOVA tested differences across multi-category groups. Pearson correlation coefficients were computed for continuous independent variables. Statistical significance was denoted by asterisks (*), with increasing asterisk count indicating stronger levels of association.

## Regression modeling and inference

To examine the relationship between environmental and demographic factors and age at menarche, we used Ordinary Least Squares (OLS) regression. While mixed-effects or longitudinal models are often suitable for repeated measures or nested data, the DHS samples are independent cross-sections with no repeated individuals or clusters. Hence, OLS was appropriate and allowed for straightforward interpretation of long-term associations across two distinct time periods.

Multicollinearity was assessed using the Variance Inflation Factor (VIF), and no significant issues were found. However, White's test revealed the presence of heteroscedasticity in both the 1992 and 2019 datasets ($p < 0.001$). Additionally, diagnostic results from Supplement text confirmed non-normality in the residuals, as both skewness and kurtosis tests were statistically significant ($p < 0.001$). Cameron & Trivedi's Information Matrix (IM) test further supported these violations, with highly significant overall chi$^2$ statistics, indicating that OLS assumptions were not met. To address this violation of OLS assumptions, a suggestion from [26] was followed. For this, robust standard errors were applied to all models to ensure accurate statistical inference. Full diagnostic test results and robustness checks are reported in S1 Text.

Regression results are presented as odds ratios with 95% confidence intervals. Significance levels are visually indicated with asterisks (*), providing a quick reference to the strength of associations. While robust analytical steps were taken, some limitations remain. Most notably, only two time points were available, which restricts the ability to model continuous changes over time; intermediate survey rounds such as NFHS-2 (2005–06) were not included due to data inconsistency and incompatibility with the merged environmental variables. Additionally, recall bias is a possible concern when relying on self-reported age at menarche. However, the DHS uses standardized questions, trained interviewers, and validated survey protocols to minimize this risk. Despite these limitations, the analysis provides a meaningful exploration of long-term changes in menarche timing across India and offers valuable insights into how environmental and demographic factors may influence adolescent reproductive health.

## Ethics statement

Since we used DHS and NASA power datasets, and both datasets were available publicly, we didn't require any additional permission to use them in our study. Both datasets are open source and can be used for various purposes in the study. Both of their data (DHS) [20,21] and the NASA power project [27] were cited in our study references, and for more information, their data policy can be found here [28,29]. So, no ethical approval was mandatory to conduct this study.

## Result

In our study, Fig 1 the bar chart depicting the mean age of first menstruation across various states of India demonstrates subtle changes between 1992 and 2019. This period marks a generally stable or slightly declining change in the age at which girls experience menarche, with the exception of Maharashtra. In most states, the mean age at menarche has either decreased marginally or remained almost consistent over the years. However, Maharashtra presents a unique case where the mean age of menarche has actually increased from 13.42 years in 1992 to 13.82 years in 2019. Given the overall change, this slight rise may seem counterintuitive, but a range of factors specific to the region could influence it.

### State-wise annual specific humidity shift

The bar charts of Fig 2 analysis of annual specific humidity at 2 meters from 1992 to 2019 reveals a uniform increase across all Indian states. This consistent rise in specific humidity levels is evident from the data over the 27 years, illustrating a clear upward trend in atmospheric moisture across diverse regions, as shown in the bar chart below. Each state demonstrated a measurable increase in humidity levels, consistent with the regional atmospheric changes, which broader climatic conditions may influence.

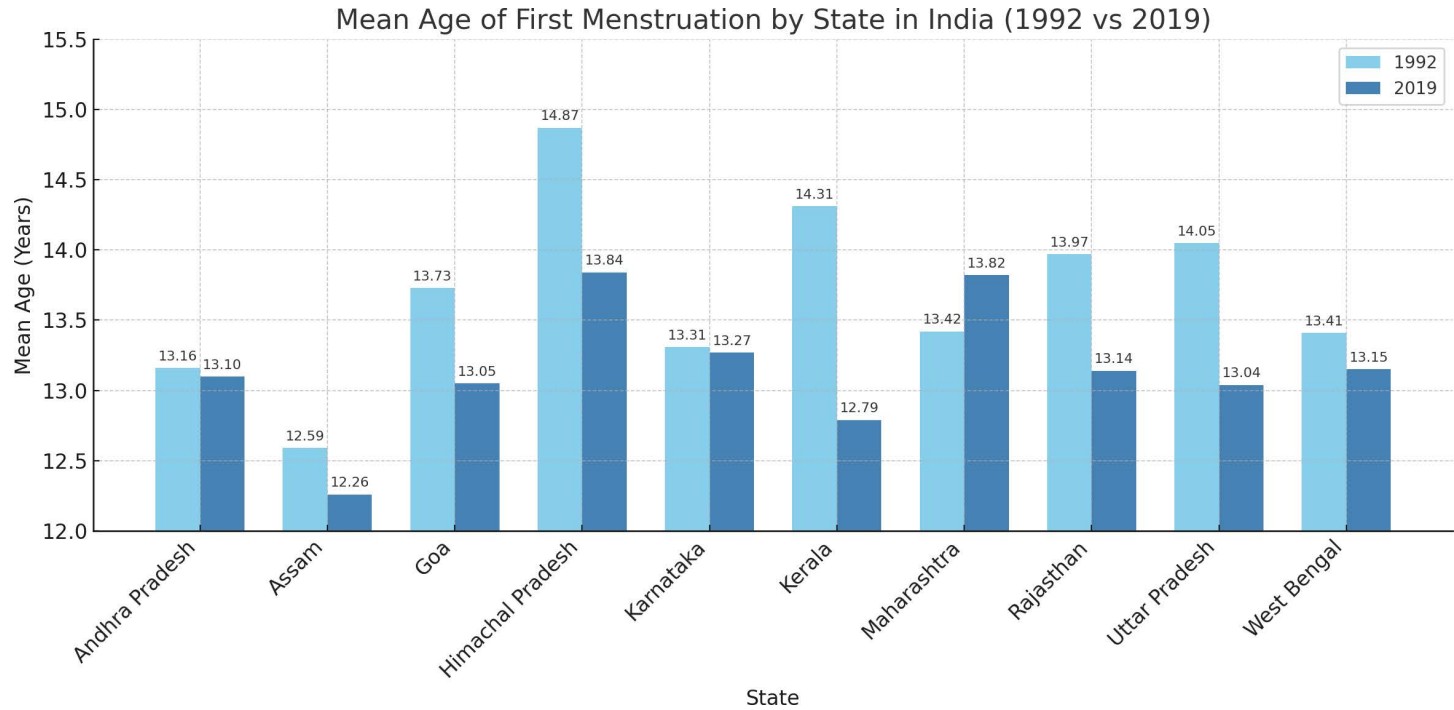

**Fig 1. Bar chart showing the mean age at first menstruation across Indian states for the years 1992–93 and 2019–21.**

### State-wise annual temperature shift

The findings of the state-wise annual temperature changes in Fig 3, as illustrated in Fig 3 through bar charts, capture subtle yet noteworthy changes in temperature profiles from 1992 to 2019 across various regions of India. The data indicate a general stability or slight upward shift in temperatures in most states, which is consistent with the broader changes of global temperature increases. However, two exceptions to this change were observed. Himachal Pradesh and Uttar Pradesh both showed a slight decrease in annual temperatures over the 27-year period. Specifically, Himachal Pradesh recorded a decrease from 20.65 °C in 1992 to 20.23 °C in 2019, while Uttar Pradesh noted a decrease from 26.39 °C in 1992 to 25.97 °C in 2019. These minor declines contrast with the general upward temperature changes observed in other states, such as Andhra Pradesh, Kerala, etc., where the temperature slightly increased during the same time frame.

Table 1 provides a comparative overview of demographic characteristics and environmental changes in India between the years 1992 and 2019. There was a significant reduction in the average age of respondents, from 20.96±8.81 years in 1992 to 19.52±2.88 years in 2019. Environmental data indicate stability in mean annual temperature at 2 meters, with a negligible increase from 25.514±1.98°C in 1992 to 25.517±1.71°C in 2019. Conversely, a substantial increase was observed in the mean annual specific humidity, rising from 11.749±2.88 in 1992 to 14.537±1.56 in 2019, highlighting a significant moistening of the climate.

The demographic distribution between urban and rural residents remained relatively unchanged. However, there was a dramatic shift in educational attainment levels over the 27-year period. The proportion of respondents with no education markedly decreased from 55.62% in 1992 to 3.63% in 2019. Correspondingly, there was an increase in respondents with secondary education from 21.15% to 72.34% and those with higher education from 3.88% to 19.15%.

Statistical analysis further explored the associations between demographic factors and the age of first menstruation. Bivariate analyses indicated significant correlations between the age of first menstruation and several independent

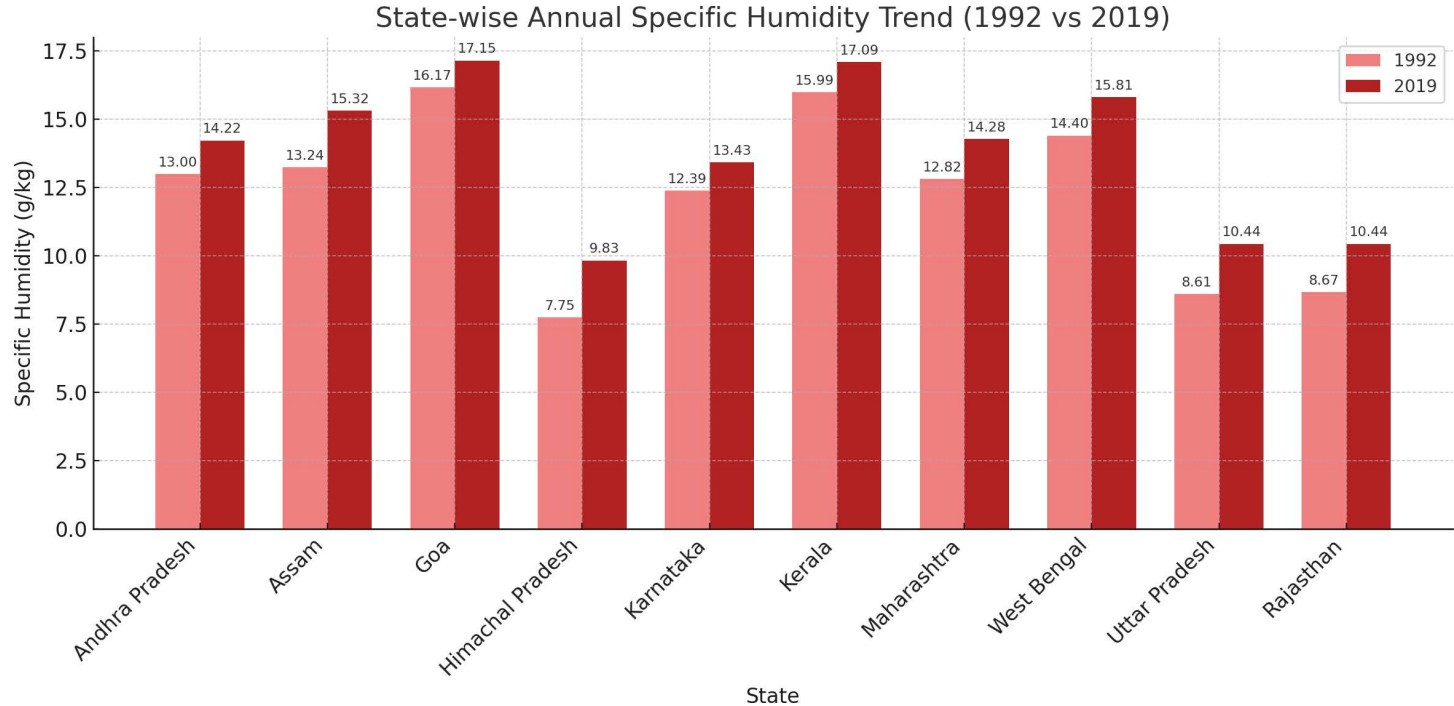

**Fig 2. Bar chart of state-wise annual specific humidity shift.**

variables, such as the respondent's current age, annual temperature at 2 meters, and annual specific humidity at 2 meters, across both 1992 and 2019 datasets. These findings were consistently supported by Pearson correlation coefficients, indicating high statistical significance. Additionally, both t-tests and ANOVA confirmed significant associations between the age of first menstruation and other variables, including residence, education level, religion, and ethnicity. These results suggest that demographic factors, alongside climatic conditions are closely linked with physiological milestones, such as the age of first menstruation, in the Indian population.

### Environmental and demographic influences on age at first menstruation

Table 2 details the findings from robust ordinary least squares (OLS) regression models that investigate the influence of various environmental and demographic characteristics on the age at first menstruation in India across different states for the years 1992 and 2019. These models provide valuable insights into the changing relationships between these characteristics and the timing of menarche over time.

### Environmental factors

The regression coefficients for annual temperature at 2 meters reveal a significant shift in its relationship with the age at first menstruation from 1992 to 2019. In 1992, higher temperatures were negatively associated with the age at first menstruation (Coefficient = -0.095, 95% CI: -0.106 to -0.085). However, by 2019, this association had reversed (Coefficient = 0.231, 95% CI: 0.222 to 0.240), indicating that higher temperatures were linked to a later onset of menstruation. Similarly, annual specific humidity at 2 meters consistently showed a negative association with the age at first menstruation for both years, with the effect size intensifying by 2019 (1992: Coefficient = -0.106, 95% CI: -0.113 to -0.098; 2019: Coefficient = -0.334, 95% CI: -0.343 to -0.323), showing increasingly earlier menstruation onset in more humid conditions.

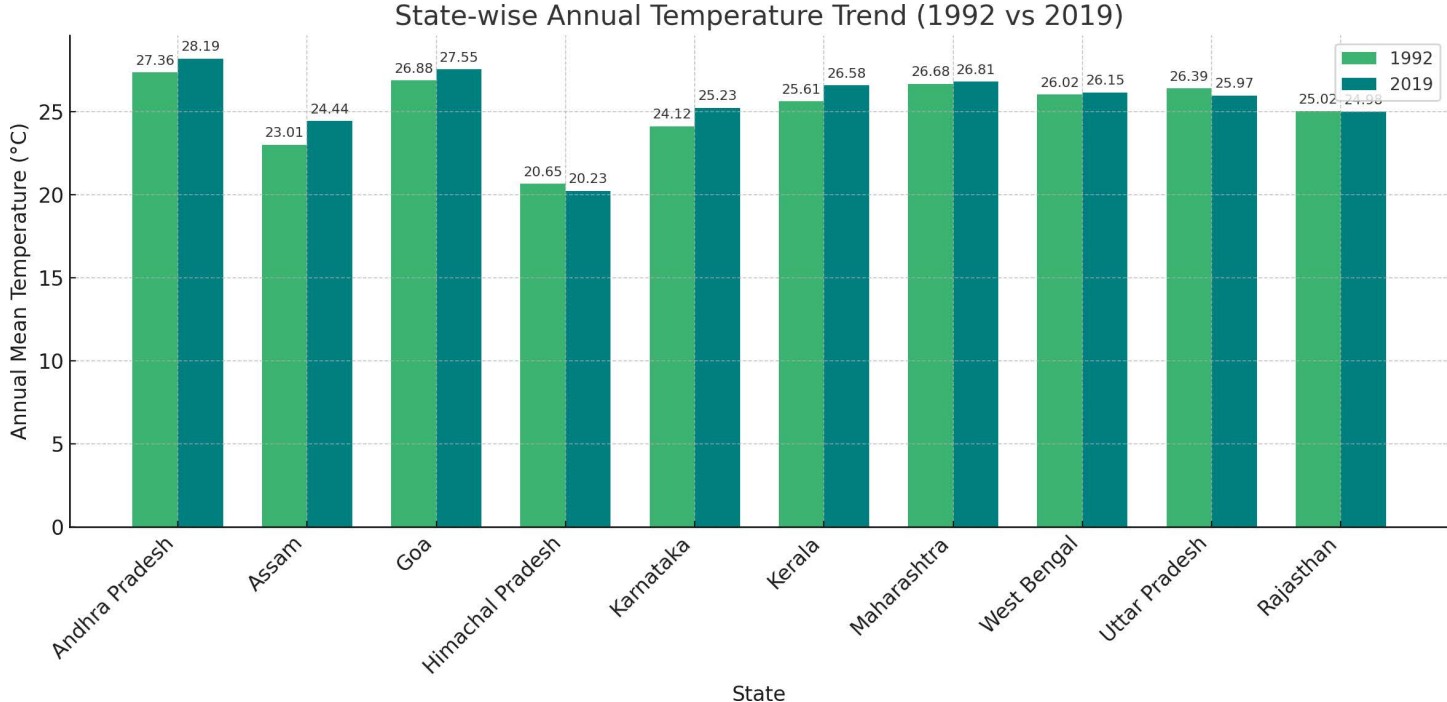

**Fig 3. Bar chart of state-wise annual temperature shift.**

### Demographic factors

The influence of residence showed notable changes over the study period. In 1992, residing in rural areas was moderately associated with a later age at first menstruation compared to urban areas (Coefficient = 0.213, 95% CI: 0.167 to 0.260). By 2019, this association had significantly weakened (Coefficient = 0.030, 95% CI: -0.003 to 0.057), reflecting a diminishing rural-urban disparity in the timing of menarche.

### Educational attainment

The association of education level with age at first menstruation also showed dynamic shifts. Those with no education served as the reference category. In 1992, all higher educational levels were positively associated with a later onset of menstruation, with secondary education showing the most substantial effect (Coefficient = 0.457, 95% CI: 0.405 to 0.509). By 2019, while the positive association persisted in secondary Coefficient = 0.186, 95%, CI: 0.119 - 0.254) higher Coefficient = 0.421, 95% CI: 0.349 - 0.493) with high statistical significance, its magnitude had decreased, particularly at the primary education level, which bordered on statistical insignificance (Coefficient = 0.042, 95% CI: -0.042 to 0.126).

### Religious and ethnic influences

Analysis of religious affiliations indicates that in 1992, Muslim and Sikh females had a significantly earlier onset of menstruation compared to Hindus, the reference group, with Muslims exhibiting a more robust effect (Coefficient for Muslims = -0.301, 95% CI: -0.355 to -0.247; Coefficient for Sikhs = -0.228, 95% CI: -0.419 to -0.037). Christian females experienced a significantly later onset (Coefficient = 0.571, 95% CI: 0.455 to 0.686). By 2019, the change for Christians reversed, showing a significantly earlier onset of menstruation (Coefficient = -0.173, 95% CI: -0.237 to -0.109). In terms of ethnicity, individuals from non-caste and non-tribe backgrounds in 1992 had a later onset of menstruation (Coefficient = 0.251, 95%

**Table 1. Demographic characteristics of the participants.**

| India 1992 | | | | India 2019 | | |
|---|---|---|---|---|---|---|
| **Characteristics** | Observation | Mean | *P-value* | Observation | Mean | *P-value* |
| **Respondent Age** | 23083 | 20.962 ± 8.81 | *** | 45329 | 19.517 ± 2.88 | *** |
| **Annual Temperature at 2 Meters** | 23083 | 25.514 ± 1.98 | *** | 45329 | 25.517 ± 1.71 | *** |
| **Annual Specific Humidity at 2 Meters** | 23083 | 11.749 ± 2.88 | *** | 45329 | 14.537 ± 1.56 | *** |
| | Freq. | Perc. % | | Freq. | Perc. % | |
| **Resident** | | | ** | | | *** |
| Urban | 5366 | 23.25 | | 11216 | 24.74 | |
| Rural | 17717 | 76.75 | | 34113 | 75.26 | |
| **Education** | | | *** | | | *** |
| No education | 12839 | 55.62 | | 1644 | 3.63 | |
| Primary | 4467 | 19.35 | | 2215 | 4.89 | |
| Secondary | 4881 | 21.15 | | 32790 | 72.34 | |
| Higher | 896 | 3.88 | | 8680 | 19.15 | |
| **Religion** | | | *** | | | *** |
| Hindu | 19159 | 83.00 | | 33329 | 73.53 | |
| Muslim | 2655 | 11.50 | | 9580 | 21.13 | |
| Christian | 814 | 3.53 | | 1545 | 3.41 | |
| Sikh | 126 | 0.55 | | 56 | 0.12 | |
| Others | 329 | 1.43 | | 819 | 1.81 | |
| **Ethnicity** | | | *** | | | *** |
| Caste | 2957 | 12.81 | | 35029 | 77.28 | |
| Tribe | 1127 | 4.88 | | 3674 | 8.11 | |
| Others | 18999 | 82/31 | | 6626 | 14.62 | |

Denote: (Significance: * p-value<0.05, ** p-value<0.01, *** p-value<0.001).

CI: 0.199 to 0.304). Still, this change was reversed by 2019 to an earlier onset (Coefficient = -0.065, 95% CI: -0.105 to -0.025), and all these are statistically significant.

## Discussion

The findings of this study provide a nuanced understanding of the factors influencing the age of menarche in various Indian states over 27 years, from 1992 to 2019. The results reveal a generally stable or slightly declining change in the age at which girls experience menarche, with the notable exception of Maharashtra, where an increase in the mean age of menarche was observed. This period marks significant demographic and environmental changes that need to be comprehensively analyzed to understand their impact on menarche changes.

The observed stability or slight decline in menarche age across most states aligns with global changes where improved nutrition, healthcare, and living standards lead to earlier menarche [8]. However, Maharashtra's increase from 13.42 years in 1992 to 13.82 years in 2019 is an exception, potentially due to regional dietary patterns, lifestyle changes, and slower socioeconomic development. According to NFHS-5 (2019–21), Maharashtra continues to report higher rates of adolescent undernutrition compared to national averages, with 35.2% of girls aged 15–19 underweight [30]. This could delay the onset of menarche, as nutritional deficiency, particularly in iron and caloric intake, has been closely linked to delayed pubertal milestones. Additionally, disparities in intrastate development and persistent rural poverty may have offset gains seen in other Indian states [31]. Socioeconomic adversity, poor nutrition, and higher physical activity levels, which delay menarche, may be more pronounced in Maharashtra [8,32,33]. Additionally, rising specific humidity generally advances

**Table 2. Robust Ordinary Least Squares (OLS) regression models examining the influence of environmental and demographic factors on the age at menarche in India for the years 1992 and 2019.**

| Characteristics | India 1992 | | India 2019 | |
|---|---|---|---|---|
| | Coefficients | 95% confidence interval | Coefficients | 95% confidence interval |
| Annual Temperature at 2 Meters | -0.095 *** | (-0.106) - (-0.085) | 0.231 *** | (0.222) - (0.240) |
| Annual Specific Humidity at 2 Meters | -0.106 *** | (-0.113) - (-0.098) | -0.334 *** | (-0.343) - (-0.323) |
| **Resident** | | | | |
| Urban | Ref | | Ref | |
| Rural | 0.213 *** | (0.167) - (0.260) | 0.030 * | (-0.003) - (0.057) |
| **Education** | | | | |
| No education | Ref | | Ref | |
| Primary | 0.331 *** | (0.282) - (0.381) | 0.042 | (-0.042) - (0.126) |
| Secondary | 0.457 *** | (0.405) - (0.509) | 0.186 *** | (0.119) - (0.254) |
| Higher | 0.177 *** | (0.080) - (0.274) | 0.421 *** | (0.349) - (0.493) |
| **Religion** | | | | |
| Hindu | Ref | | Ref | |
| Muslim | -0.301 *** | (-0.355) - (-0.247) | -0.008 | (-0.041) - (0.025) |
| Christian | 0.571 *** | (0.455) - (0.686) | -0.173 *** | (-0.237) - (-0.109) |
| Sikh | -0.228 * | (-0.419) - (-0.037) | 0.239 | (-0.150) - (0.630) |
| Other | 0.158 * | (-0.005) - (0.311) | 0.254 *** | (0.163) - (0.344) |
| **Ethnicity** | | | | |
| **Caste** | Ref | | Ref | |
| Tribe | -0.017 | (-0.103) - (0.069) | 0.006 | (-0.036) - (0.050) |
| Other | 0.251 *** | (0.199) - (0.304) | -0.065 ** | (-0.105) - (-0.025) |

Denote: (P-value indication: * p-value<0.05, ** p-value<0.01, *** p-value<0.001).

menarche by altering underlying health conditions. However, the relationship between higher temperatures and delayed menarche observed in Maharashtra may reflect physiological stress due to chronic heat exposure, which can impact hormonal regulation and delay puberty. This anomaly might also be influenced by local environmental factors such as frequent droughts, water scarcity, or seasonal migration patterns that affect nutrition, psychosocial stress, and healthcare access [34,35]. To address these challenges, public health programs in Maharashtra should strengthen adolescent-focused nutrition interventions through schemes like the Integrated Child Development Services (ICDS) and Mid-Day Meal programs.

Analyzing environmental influences, the study highlights a uniform increase in specific humidity across all states, indicating an increasingly moist climate over the study period. The negative association between specific humidity and the age of menarche, which intensified by 2019, suggests that higher humidity levels might be contributing to earlier menarche. This is consistent with findings that improved health conditions, which often accompany higher humidity levels, can accelerate biological maturation processes [34]. Moreover, the interaction of humidity and temperature may influence pubertal timing through thermoregulatory and endocrine pathways. While higher humidity supports hydration and metabolic stability, extreme heat exposure—as noted in Maharashtra—can activate stress responses that suppress the hypothalamic–pituitary–gonadal (HPG) axis, delaying hormonal signaling involved in puberty onset [36,37]. These physiological effects may be compounded by chronic undernutrition, drought-related food insecurity, and high physical exertion among adolescents in rural regions. Maharashtra's divergence, therefore, likely reflects a convergence of climatic stressors and socioeconomic vulnerabilities, including migration patterns that disrupt consistent nutrition and

healthcare access. Together, these mechanisms may help explain the state's delayed menarche trend despite broader national advances.

Temperature shifts varied across states, with most experiencing slight increases in annual temperatures, except for Himachal Pradesh and Uttar Pradesh, where decreases were noted. Warmer temperatures can influence the hormonal regulation involved in puberty, potentially delaying menarche if the body's adaptive responses prioritize energy expenditure for thermoregulation over reproductive development [38,39].

Demographic changes, particularly in educational attainment, also played a significant role in influencing menarche change. The sharp decrease in the proportion of respondents with no education, coupled with an increase in those with secondary and higher education, underscores significant improvements in educational access. Higher education levels have been consistently linked to earlier menarche, likely due to better health awareness and access to nutritional resources [40]. This finding may appear counterintuitive compared to global trends, where higher education is often linked with delayed menarche due to lower obesity rates and increased physical activity. However, in India, especially in rural contexts, higher educational attainment can signal better access to school-based health programs, midday meal schemes, and improved sanitation—all of which can enhance nutritional and physical health and potentially lead to earlier puberty. Moreover, educational level may reflect broader household wealth in these regions, indirectly correlating with better health indicators [41,42]. This paradoxical relationship between higher education and earlier menarche in the Indian context may also reflect unique behavioral and environmental factors. Increased academic engagement might reduce physical activity levels or lead to sedentary routines, which have been associated with earlier puberty. Additionally, school-based meal programs could improve caloric and micronutrient intake, particularly in rural settings, accelerating physiological development [43]. Together, these mechanisms may explain the deviation from global patterns and underscore the need to interpret education effects within specific cultural and nutritional contexts. However, the weakening of this association by 2019 suggests that other factors, such as urbanization and overall improvements in living conditions, might also be crucial determinants [40].

Religious and ethnic influences on menarche age also demonstrated notable shifts. For example, in 1992, Muslim and Sikh girls experienced earlier menarche compared to Hindus, while by 2019, Christian girls, who initially experienced later menarche, showed an earlier onset. These changes may reflect broader sociocultural transformations and improved healthcare access across different communities, reducing disparities observed in earlier decades [44]. The evolving socio-religious landscape in India, with increasing inter-community interactions and shared health practices, could also contribute to these changes.

The bivariate analyses indicated significant correlations between the age of first menstruation and several independent variables, including the respondent's current age, annual temperature at 2 meters, and annual specific humidity at 2 meters across both 1992 and 2019 datasets. Pearson correlation coefficients consistently supported these findings, indicating high statistical significance. Additionally, ANOVA confirmed significant associations between the age of first menstruation and other continuous variables. For the categorical variable $chi^2$, the association included residence, education level, religion, and ethnicity. These results suggest that demographic factors, alongside climatic conditions, are closely linked with physiological milestones such as the age of first menstruation in the Indian population.

The multivariate analysis, specifically the robust ordinary least squares (OLS) regression models, provided valuable insights into the changing relationships between these characteristics and the timing of menarche over time. The regression coefficients for annual temperature at 2 meters revealed a significant shift in its relationship with the age at first menstruation from 1992 to 2019. In 1992, higher temperatures were negatively associated with the age at first menstruation, indicating that higher temperatures were linked to an earlier onset of menstruation. By 2019, this association had reversed, showing that higher temperatures were linked to a later onset of menstruation. This reversal may reflect chronic heat stress adaptation, where prolonged exposure to elevated temperatures disrupts the hypothalamic-pituitary-gonadal axis, delaying hormonal maturation. Emerging evidence suggests that sustained thermal load may alter leptin sensitivity

and suppress gonadotropin-releasing hormone (GnRH) secretion, thereby postponing the onset of puberty. Such biological stress responses could become more pronounced as climate extremes intensify [45,46] implementing heat-resilient healthcare infrastructure and localized climate adaptation measures can support hormonal and physical development in high-risk regions [47]. Similarly, annual specific humidity at 2 meters consistently showed a negative association with the age at first menstruation for both years, with the effect size intensifying by 2019, indicating increasingly earlier menstruation onset in more humid conditions.

The influence of residence showed notable changes over the study period. In 1992, residing in rural areas was moderately associated with a later age at first menstruation compared to urban areas, likely due to differences in nutrition, healthcare access, and overall living conditions between rural and urban settings [48]. By 2019, this association had significantly weakened, reflecting a diminishing rural-urban disparity in the timing of menarche. This could be attributed to the overall improvement in living standards and healthcare access in rural areas, reducing the gap between urban and rural regions [49].

The association of education level with age at first menstruation also showed dynamic shifts. Those with no education served as the reference category. In 1992, all higher educational levels were positively associated with a later onset of menstruation, with secondary education showing the most substantial effect. This association is supported by previous literature indicating that higher education levels are linked to better health awareness and access to nutritional resources, which can delay menarche [50]. By 2019, while the positive association persisted, its magnitude had decreased, particularly at the primary education level, which bordered on statistical insignificance. This decrease might be due to overall improvements in nutritional status and health awareness across all education levels, narrowing the differences [51].

Analysis of religious affiliations indicates that in 1992, Muslim and Sikh females had a significantly earlier onset of menstruation compared to Hindus, the reference group, with Muslims exhibiting a more robust effect. Christian females experienced a significantly later onset, which aligns with previous findings indicating variations in menarche age across different religious groups due to sociocultural and nutritional differences [52]. By 2019, the change for Christians reversed, showing a significantly earlier onset of menstruation. This shift could reflect improved healthcare access and nutritional status among Christian communities over the years, aligning their menarche age more closely with other groups [52].

In terms of ethnicity, individuals from non-caste and non-tribe backgrounds in 1992 had a later onset of menstruation, but this change reversed by 2019 to an earlier onset, and all these changes were statistically significant. This shift might indicate broader sociocultural changes and improved access to healthcare and nutrition across different ethnic groups, reducing disparities in menarche timing [53]. The detailed analysis of environmental and demographic factors influencing menarche age in this study underscores the complex interplay of these variables. The findings highlight the importance of considering regional specificities and broader socioeconomic changes when analyzing physiological milestones like menarche. These findings also align with India's National Action Plan on Climate Change (NAPCC), particularly under the National Mission on Health and the National Mission on Strategic Knowledge for Climate Change [54]. Integrating adolescent health indicators such as the timing of menarche into climate adaptation and health policy planning is essential to ensure that public health responses are climate-resilient, inclusive, and tailored to the vulnerabilities of young populations.

## Limitation

While this study employed a thorough methodology, one primary limitation is the reliance on self-reported information from DHS surveys, which introduces potential biases. Factors such as cultural perceptions and recall errors—especially for retrospective variables like age at menarche—may compromise the accuracy of responses. Another notable limitation arises from the mismatch in the timing of climate exposure and menarche occurrence. As the DHS dataset does not provide the precise month or season of menarche, and given the retrospective nature of reporting, we used annual climate averages from NASA's POWER database to align with DHS geographic units. However, this temporal misalignment, combined with

the use of annual rather than seasonal data, may obscure short-term climatic influences such as seasonal extremes or acute weather events, limiting precision in interpreting climate–menarche associations.

Additionally, this study is subject to the inherent constraints of using secondary data. Key biological and physiological variables—such as BMI, hormonal profiles, heredity, and physical activity—were not available in either DHS or NASA datasets, and therefore could not be included in our models. While BMI is a particularly relevant confounder, harmonized BMI data was not consistently available across both survey years (1992–93 and 2019–21). As a result, we employed proxy indicators like place of residence, education, and ethnicity, which are known correlates of nutritional and health status in the Indian context. Nonetheless, the absence of direct biological measures limits our ability to fully disentangle environmental from biological effects on menarche timing. Geographic representativeness also poses limitations. The exclusion of certain states, such as Jammu & Kashmir and Punjab, due to incomplete or incompatible data, may introduce selection bias and restrict the generalizability of the findings to all Indian regions. Moreover, the contextual variability between 1992 and 2019—including socioeconomic shifts, policy changes, and regional crises—introduces unmeasured variation that could confound the observed associations. Finally, as a cross-sectional analysis, this study cannot infer causality or mediation pathways. Future research would benefit from longitudinal or cohort-based designs incorporating biological markers, seasonally resolved climate exposures, and expanded geographic coverage to validate and expand upon these findings.

## Conclusion

This study explored environmental and demographic associations with age at menarche across Indian states using DHS data from 1992 and 2019, revealing a slight decline in menarcheal age overall, with Maharashtra as an exception. While demographic factors remain key determinants, environmental variables—particularly humidity and temperature—also appear linked to menarche timing. These findings suggest the need for community-level interventions that enhance adolescent nutrition, ensure accessible reproductive healthcare, and promote awareness of climate-related health risks. In climate-vulnerable regions, school-based health programs, early adolescent health monitoring, and localized awareness campaigns may help buffer environmental impacts on reproductive development. Public health strategies should prioritize context-specific adaptation and continuous monitoring of climate-health linkages to safeguard adolescent well-being.

## Supporting information

**S1 Text. Overall IM test.**
(DOCX)

## Acknowledgments

The authors thank the Demographic and Health Survey and the NASA Power Project for keeping their data open-source to utilize them in research work. Without these, it would not be possible to complete this manuscript.

## Author contributions

**Conceptualization:** MD. Nahid Hassan Nishan.

**Data curation:** MD. Nahid Hassan Nishan.

**Formal analysis:** MD. Nahid Hassan Nishan, Anika Ferdous, M. Z. E. M. Naser Uddin Ahmed.

**Methodology:** MD. Nahid Hassan Nishan.

**Software:** MD. Nahid Hassan Nishan.

**Visualization:** MD. Nahid Hassan Nishan.

**Writing – original draft:** MD. Nahid Hassan Nishan, Anika Ferdous, M. Z. E. M. Naser Uddin Ahmed, Khadiza Akter.

**Writing – review & editing:** MD. Nahid Hassan Nishan, Anika Ferdous, M. Z. E. M. Naser Uddin Ahmed, Khadiza Akter.

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
