## [Decision Letter · Decision Letter 0]

5 May 2025

PGPH-D-25-00529

Shifts in Menarche Age: Environmental and Demographic Influences Over a Quarter Century in India

Dear Dr. NISHAN,

Thank you for submitting your manuscript to PLOS Global Public Health. After careful consideration, we feel that it has merit but does not fully meet PLOS Global Public Health’s publication criteria as it currently stands. Therefore, we invite you to submit a revised version of the manuscript that addresses the points raised during the review process.

We look forward to receiving your revised manuscript.

Kind regards,

Jianhong Zhou

Staff Editor

Journal Requirements:

1. We suggest you thoroughly copyedit your manuscript for language usage, spelling, and grammar. If you do not know anyone who can help you do this, you may wish to consider employing a professional scientific editing service.

Additional Editor Comments (if provided):

Reviewers' comments:

Reviewer's Responses to Questions

**Comments to the Author**

1. Does this manuscript meet PLOS Global Public Health’s publication criteria ? Is the manuscript technically sound, and do the data support the conclusions? The manuscript must describe methodologically and ethically rigorous research with conclusions that are appropriately drawn based on the data presented.

Reviewer #1: No

Reviewer #2: Yes

2. Has the statistical analysis been performed appropriately and rigorously?

Reviewer #1: No

Reviewer #2: Yes

3. Have the authors made all data underlying the findings in their manuscript fully available (please refer to the Data Availability Statement at the start of the manuscript PDF file)?

Reviewer #1: No

Reviewer #2: Yes

4. Is the manuscript presented in an intelligible fashion and written in standard English?

Reviewer #1: No

Reviewer #2: Yes

5. Review Comments to the Author

Reviewer #1: The main factors affecting Menarche:

1. Heredity

2. Hormones

3. Amount of fat in the body

4. Health condition

5. Physical exercise

None of these factors were used as controls for climate factors. If only demographic variables are used as controls, it is not appropriate to see the role of climate.

The DHS data uses a cross-sectional research design so it is not possible to see the effect. The purpose of the analysis is to see the relationship between the dependent variable and the independent variable

There is no explanation of the data management process for data coming from several sources

Is the availability of regional data in climate data the same as DHS data? Explain

What is the timing of climate observations and individual menarche times. Has it been adjusted for each individual to provide an explanation of the relationship between menarche and climate?

Reviewer #2: 1. The first sentence of the Introduction section needs to be rewritten.

2. Reference 6, which points to the trend of menarche age in China, is too old and unsuitable for mentioning and comparing. Please find newer studies and refer to them.

3. Please write the Introduction section in 4 or 5 separate paragraphs, but link the contents with related sentences.

Results section:

Simplify the data analysis subsection by condensing technical details (e.g., heteroscedasticity correction) into a separate supplementary section.

Ensure figures (e.g., bar charts for state-wise trends) are clearly labeled and visually accessible. For example, Figure 1’s y-axis should explicitly state "Mean Age (Years)" to avoid ambiguity.

Discussion

• provide specific socioeconomic or dietary data (e.g., NFHS reports, state-level nutrition surveys) to contextualize why Maharashtra diverges from national trends.

• Discuss potential biological or environmental mechanisms (e.g., heat stress adaptation, hormonal changes) for the shift from a negative (1992) to positive (2019) temperature effect.

• Clarify why higher education correlates with earlier menarche despite global trends linking education to delayed menarche. Consider regional factors like urban-rural disparities in educational quality or health access.

Statistical Analysis

• Justify the use of OLS over mixed-effects or longitudinal models, given the cross-sectional, time-segmented design.

• Address limitations of using only two time points (1992 and 2019) by discussing whether intermediate trends (e.g., 2005-06 NFHS data) were explored or could refine results.

Limitations

• Expand on potential recall bias mitigation strategies (e.g., DHS survey protocols, validation studies).

• Acknowledge that excluding states due to data gaps (e.g., northern regions like Punjab) may limit national representativeness.

Policy Implications

• Specify how public health programs in Maharashtra could address delayed menarche (e.g., nutrition supplementation, heat-resilient healthcare infrastructure).

• Link findings to India’s climate action plans (e.g., National Action Plan on Climate Change) to emphasize urgency.

• Revise fragmented sentences (e.g., "We used India's Demographic and Health Survey dataset. (DHS) contains...") and reduce redundancy in the methodology.

• Verify that all in-text citations (e.g., [20] for latitude/longitude sources) are correctly formatted and accessible.

6. PLOS authors have the option to publish the peer review history of their article (what does this mean? ). If published, this will include your full peer review and any attached files.

**Do you want your identity to be public for this peer review?** For information about this choice, including consent withdrawal, please see our Privacy Policy .

Reviewer #1: No

Reviewer #2: No

---

## [Decision Letter · Decision Letter 1]

10 Jun 2025

PGPH-D-25-00529R1

Shifts in Menarche Age: Environmental and Demographic Influences Over a Quarter Century in India

Dear Dr. NISHAN,

Thank you for submitting your manuscript to PLOS Global Public Health. After careful consideration, we feel that it has merit but does not fully meet PLOS Global Public Health’s publication criteria as it currently stands. Therefore, we invite you to submit a revised version of the manuscript that addresses the points raised during the review process.

In your revision please pay attention to the remaining comments from Reviewer #1. If point #3 has already been addressed in this revision, please describe that more carefully in your response. I believe that Points #4 and #7 are suggestions and I think you could choose to provide more clarity about your decision making process, rather than revise the analysis. 

We look forward to receiving your revised manuscript.

Kind regards,

Nicola Hawley

Academic Editor

Journal Requirements:

1. We would like to request copy editing.

Additional Editor Comments (if provided):

Reviewers' comments:

Reviewer's Responses to Questions

**Comments to the Author**

1. If the authors have adequately addressed your comments raised in a previous round of review and you feel that this manuscript is now acceptable for publication, you may indicate that here to bypass the “Comments to the Author” section, enter your conflict of interest statement in the “Confidential to Editor” section, and submit your "Accept" recommendation.

Reviewer #1: (No Response)

Reviewer #2: All comments have been addressed

2. Does this manuscript meet PLOS Global Public Health’s publication criteria ? Is the manuscript technically sound, and do the data support the conclusions? The manuscript must describe methodologically and ethically rigorous research with conclusions that are appropriately drawn based on the data presented.

Reviewer #1: Partly

Reviewer #2: Yes

3. Has the statistical analysis been performed appropriately and rigorously?

Reviewer #1: No

Reviewer #2: Yes

4. Have the authors made all data underlying the findings in their manuscript fully available (please refer to the Data Availability Statement at the start of the manuscript PDF file)?

Reviewer #1: No

Reviewer #2: Yes

5. Is the manuscript presented in an intelligible fashion and written in standard English?

Reviewer #1: Yes

Reviewer #2: Yes

6. Review Comments to the Author

Reviewer #1: 1. In the introduction, the issue of menarche can be clarified. What is the ideal age for menarche? What are the effects of early or late menarche?

2. The term “trend” cannot be used because there are only two data points.

3. The use of climate data that does not align with the age of menarche, for example: a respondent who was 15 years old during the 1992 survey and reported menarche at age 13 should have used climate data from 1990 (two years prior).

4. Suggestion: if the argument is that climate conditions have not changed significantly, then both climate data should be averaged. Additionally, the climate data used for analysis is only from 2019, while the climate data uses the average of data from 1992 and 2019.

5. What is the basis for using the religion indicator as a predictor? Explain with appropriate literature support.

6. The conclusion is unclear. If climate plays a role in menarche, what solutions can be implemented by the community? Are there any interventions that can be carried out or adaptations that the community can make?

7. re-analyze so that the theory is more consistent with the data used. Another thing to consider after changing the analysis is the title of the manuscript.

Reviewer #2: (No Response)

7. PLOS authors have the option to publish the peer review history of their article (what does this mean? ). If published, this will include your full peer review and any attached files.

**Do you want your identity to be public for this peer review?** For information about this choice, including consent withdrawal, please see our Privacy Policy .

Reviewer #1: No

Reviewer #2: No

---

## [Decision Letter · Decision Letter 2]

18 Jul 2025

PGPH-D-25-00529R2

Understanding Age at Menarche: Environmental and Demographic Influences Over a Quarter Century in India

Dear Dr. NISHAN,

Thank you for submitting your manuscript to PLOS Global Public Health. After careful consideration, we feel that it has merit but does not fully meet PLOS Global Public Health’s publication criteria as it currently stands. Therefore, we invite you to submit a revised version of the manuscript that addresses the points raised during the review process.

I am grateful for the attention paid to the reviewer comments during the last revision - you will see that your edits addressed most of the reviewers major concerns. Both reviewers though, continue to raise the need to address the theoretical link between climate and menarche. Please address that comment with this revision. Reviewer #2 asks for additional clarity around validity of the measures, which I also think is important to address. 

We look forward to receiving your revised manuscript.

Kind regards,

Nicola Hawley

Academic Editor

Journal Requirements:

1.  We would like to request copy editing.

Additional Editor Comments (if provided):

Reviewers' comments:

Reviewer's Responses to Questions

**Comments to the Author**

1. If the authors have adequately addressed your comments raised in a previous round of review and you feel that this manuscript is now acceptable for publication, you may indicate that here to bypass the “Comments to the Author” section, enter your conflict of interest statement in the “Confidential to Editor” section, and submit your "Accept" recommendation.

Reviewer #1: All comments have been addressed

Reviewer #2: (No Response)

2. Does this manuscript meet PLOS Global Public Health’s publication criteria ? Is the manuscript technically sound, and do the data support the conclusions? The manuscript must describe methodologically and ethically rigorous research with conclusions that are appropriately drawn based on the data presented.

Reviewer #1: Yes

Reviewer #2: Yes

3. Has the statistical analysis been performed appropriately and rigorously?

Reviewer #1: Yes

Reviewer #2: Yes

4. Have the authors made all data underlying the findings in their manuscript fully available (please refer to the Data Availability Statement at the start of the manuscript PDF file)?

Reviewer #1: Yes

Reviewer #2: No

5. Is the manuscript presented in an intelligible fashion and written in standard English?

Reviewer #1: Yes

Reviewer #2: Yes

6. Review Comments to the Author

Reviewer #1: 1. Add a theory that explains

a) how climate affects menarche

b) at what age menarche should occur and the effects of menarche occurring too early or too late.

2. Check line 279, is the sentence complete?

3. Add limitations regarding the status of climate data that is not ideal. The years of climate data collection and menarche are not identical.

4. Explain the role of religion in differences in menarche. Do religious rules cause these differences?

Reviewer #2: Dear editor

Thank you for providing me with this opportunity to read the manuscript titled: Understanding Age at Menarche: Environmental and Demographic Influences Over a Quarter Century in India

This paper offers interesting insights into the interaction of environmental and demographic factors with menarche in India. It is strong on policy relevance and data integration but can improve on methodological transparency and sensitive interpretation.

In the Abstract:

Abbreviate the methods section (e.g., "cross-sectional analysis of DHS and NASA data" is enough).

In the introduction section, Condense the literature review; certain paragraphs (e.g., comparisons of caste) might be shortened.

In the Methodology: Self-Report Bias, although mentioned, the effect of recall bias on menarche age (retrospective data) requires more in-depth discussion.

Climate Data Granularity: Annual averages can hide seasonal and climatic extremes. Explain why multi-year averages were not used.

Missing Variables: Lack of biological markers (hormones, BMI) is a significant limitation. Propose proxy variables (e.g., DHS wealth index for nutrition).

In the Statistical presentation:

Tables/Figures: Be consistent in formatting (e.g., confidence intervals in Table 2). Report p-values with coefficients.

Supplemental Data: The test for heteroscedasticity (Supplement 1) needs to be summarized in the main text.

In the findings:

Education Paradox: The association of earlier menarche with higher education is contrary to international trends. Discuss possible mechanisms (e.g., school lunch programs vs. stress/physical activity).

Avoid repetitive statements (for example, Maharashtra's anomaly is stated more than once).

Maharashtra's Anomaly: Speculate as to why this state differs (e.g., local droughts, migration patterns).

In the discussion section:

Climate Mechanisms: Explain how humidity/temperature physiologically impact puberty (e.g., hydration, thermoregulation stress). Elaborate on mechanisms underlying crucial findings (e.g., Maharashtra's trend).

Discussed limitations more critically (e.g., biological confounders).

Ethical and Generalizability Notes: State Selection Bias: Excluded states (e.g., Punjab, Jammu & Kashmir) can distort findings. Address regional representativeness.

Open Data: Indicate whether raw/processed data are made available (e.g., through repositories such as Figshare).

In the References: Verify that all citations are consistent with the reference list (e.g., verify numbering in Discussion).

7. PLOS authors have the option to publish the peer review history of their article (what does this mean? ). If published, this will include your full peer review and any attached files.

**Do you want your identity to be public for this peer review?** For information about this choice, including consent withdrawal, please see our Privacy Policy .

Reviewer #1: No

Reviewer #2: No

---

## [Editor Report · Decision Letter 3]

11 Aug 2025

Understanding Age at Menarche: Environmental and Demographic Influences Over a Quarter Century in India

PGPH-D-25-00529R3

Dear Dr. NISHAN,

We are pleased to inform you that your manuscript 'Understanding Age at Menarche: Environmental and Demographic Influences Over a Quarter Century in India' has been provisionally accepted for publication in PLOS Global Public Health.

Best regards,

Nicola Hawley

Academic Editor